# Exploring the Structurally Conserved Regions and Functional Significance in Bacterial N-Terminal Nucleophile (Ntn) Amide-Hydrolases

**DOI:** 10.3390/ijms25136850

**Published:** 2024-06-21

**Authors:** Israel Quiroga, Juan Andrés Hernández-González, Elizabeth Bautista-Rodríguez, Alfredo C. Benítez-Rojas

**Affiliations:** 1Department of Life and Health Sciences, Universidad Popular Autónoma del Estado de Puebla, 13 Poniente No. 1927, Barrio de Santiago, Puebla 72410, Mexico; juanandres.hernandez@upaep.edu.mx (J.A.H.-G.); eli.bautista@gmail.com (E.B.-R.); alfredocesar.benitez@upaep.mx (A.C.B.-R.); 2Department of Health Sciences, Universidad Autónoma de Tlaxcala, Sur 11, Barrio de Guardia, Zacatelco 90070, Mexico

**Keywords:** N-terminal nucleophile (Ntn) amide-hydrolase, structure–function relationship, protein modeling, protein design

## Abstract

The initial adoption of penicillin as an antibiotic marked the start of exploring other compounds essential for pharmaceuticals, yet resistance to penicillins and their side effects has compromised their efficacy. The N-terminal nucleophile (Ntn) amide-hydrolases S45 family plays a key role in catalyzing amide bond hydrolysis in various compounds, including antibiotics like penicillin and cephalosporin. This study comprehensively analyzes the structural and functional traits of the bacterial N-terminal nucleophile (Ntn) amide-hydrolases S45 family, covering penicillin G acylases, cephalosporin acylases, and D-succinylase. Utilizing structural bioinformatics tools and sequence analysis, the investigation delineates structurally conserved regions (SCRs) and substrate binding site variations among these enzymes. Notably, sixteen SCRs crucial for substrate interaction are identified solely through sequence analysis, emphasizing the significance of sequence data in characterizing functionally relevant regions. These findings introduce a novel approach for identifying targets to enhance the biocatalytic properties of N-terminal nucleophile (Ntn) amide-hydrolases, while facilitating the development of more accurate three-dimensional models, particularly for enzymes lacking structural data. Overall, this research advances our understanding of structure–function relationships in bacterial N-terminal nucleophile (Ntn) amide-hydrolases, providing insights into strategies for optimizing their enzymatic capabilities.

## 1. Introduction

Penicillin, as an antibiotic, was only the beginning of the discovery of other molecules now considered indispensable drugs for humanity. However, with the emergence of resistance to penicillins and their associated side effects, their use as a first-choice antibiotic was ruled out. Various infections are becoming an increasingly common problem. Given this, the discovery of 6-aminopenicillanic acid (6-APA) led to the synthesis of semi-synthetic penicillins (ampicillin and amoxicillin) through modifying wild penicillins, which provided a promising response to bacterial resistance. Previous studies [1,2] reported that amidases or amidohydrolases are the biocatalysts responsible for the synthesis of chiral carboxylic acids, constituting a large group of enzymes capable of hydrolyzing the C-N bridges of various amide compounds, thus producing carboxylic acids. Among the excellent properties of these acylases are their broad substrate specificity, their remarkable regio-, chemo-, and enantio-selectivity, and their catalyzation without cofactors, which makes them the most versatile enzymes in the pharmaceutical industry and the chemical synthesis of basic products. Other enzymes capable of producing 6-APA, 7-ADCA (7-aminodesacetoxycephalosporanic acid), or 7-ACA (7-aminocephalosporanic acid) through the hydrolysis of penicillins or cephalosporins are called beta-lactam acylases [1].

Penicillin acylases, cephalosporin acylases, and D-succinylase belong to the family of enzymes known as N-terminal nucleophile (Ntn) amide-hydrolases [3,4], specifically to the Serine (S) Peptidases S45 family (MEROPS Accession MER0003306) [5]. These enzymes are responsible for catalyzing the hydrolysis of amide bonds in various compounds, including antibiotics like penicillin and cephalosporin, as well as D-succinylase which acts on D-amino acids [3,6,7]. A profound knowledge of the factors mediating the selectivity and activity of these proteins is a prerequisite for developing enzymes with improved properties [8,9,10]. Therefore, deeper insights into the relationships between sequence, structure, and function are of great interest [9,10,11,12]. Despite their sequence and substrate preferences’ dissimilarities, these enzymes converge on a shared structural superfamily framework, bearing the hallmark of a catalytic serine, cysteine, or threonine at the N-terminal position [13,14]. This structural unity amidst sequence variability hints at an evolutionary convergence that has shaped their functional diversity. The general 3D structure of amidases, including penicillin G acylases, cephalosporin acylases, and D-succinylase, typically involves a specific fold known as the α/β-hydrolase fold [15,16,17,18]. This fold consists of alternating alpha-helices and beta-sheets arranged in a sandwich-like structure. The genes in this protein family encode a precursor polypeptide with four structural elements: a signal peptide, α- and β-subunits, and an inter-subunit spacer [19]. The maturation of the zymogen involves the autocatalytic cleavage of the inter-subunit spacer. The mature protein forms a heterodimer consisting of α- and β-subunits. The α-subunit folds first and serves as a template for the proper folding of the β-subunit [20]. 

These enzymes often have a catalytic triad composed of three critical amino acids: a nucleophile (often a serine), an acid (usually a glutamate or aspartate), and a base (often a histidine). This catalytic triad is crucial for the enzymatic hydrolysis of amide bonds. The active site where the substrate binds and the enzymatic reaction occurs is usually located in a groove or pocket within the enzyme structure. This pocket allows for specific interactions between the enzyme and its substrate, facilitating the enzymatic reaction [21]. The amidases’ 3D structure is crucial for their function in cleaving amide bonds in various compounds, including antibiotics and other substrates. The specific arrangement of amino acids and the catalytic triad within their structure enables their enzymatic activity [13,21].

The active site architecture of penicillin acylases enzymes represents a highly specialized molecular environment crucial for catalytic activity. Within this site, a conserved arrangement of amino acid residues orchestrates the binding and subsequent hydrolysis of β-lactam antibiotics [22,23]. Structural analyses have revealed a pocket-like configuration in the enzyme’s tertiary structure, accommodating the substrate molecule with a specific conformation [24]. Key catalytic residues, such as a serine, cysteine, or threonine positioned at the N-terminus, serve as nucleophiles, initiating the amide bond cleavage in the β-lactam ring [25]. Surrounding these catalytic residues are amino acid residues that contribute to substrate recognition, binding, and stabilization. Notably, the active site comprises a co-ordinated arrangement of amino acids and solvent molecules that optimize interactions with the substrate, enabling the precise and selective hydrolytic action characteristic of penicillin acylases [26,27]. The structural intricacies and specific spatial arrangement of residues within this active site environment underscore its pivotal role in the enzyme’s catalytic function and its significance in antibiotic biotransformation processes. In illuminating the multifaceted nature of these enzymes and their profound implications in drug development and biotechnological innovation, this article aims to guide future scientific inquiry, inspiring breakthroughs that could redefine the boundaries of pharmaceutical and biotechnological advancements. The highly similar structures have been compared in detail to identify the common core and to assign the variable regions. For this purpose, a structural alignment was used as a base to generate a reliable structure profile. With this profile, all structurally conserved regions (SCRs) could be predicted and annotated among these proteins, hence allowing a structural navigation in their sequences lacking structural information. This critical approach holds the potential to unveil the intricacies underlying their functional diversity and to provide a roadmap for understanding the relationship between their sequences and structural motifs.

In seeking to reconcile the disparity between sequence variance and structural preservation, this article harnesses the capabilities of structural bioinformatics tools, thereby laying the groundwork for a more profound understanding of these enzymes’ evolution, functionality, and prospective customized modifications. Through this exploration, it is anticipated that a clearer understanding of the factors dictating selectivity and activity will emerge, facilitating the rational design of enzymes with enhanced properties for diverse applications. Therefore, this article delves into the complex landscape of penicillin G acylases, cephalosporin acylases, and D-succinylase, merging structural insights with functional implications to elucidate aspects that enable the hypothesis of innovative trajectories in drug development and biotechnology.

## 2. Results

### 2.1. Structural Analysis

Through a comprehensive comparative analysis, we have discerned distinct structural features among the N-terminal nucleophile (Ntn) amide-hydrolases S45 family. The calculation of the root-mean-square deviation (RMSD in the SALIGN server and VMD) and structural comparison across the 83 crystal structures archived in the Protein Data Bank (PDB) unveiled ten structural variants. These 83 structures were meticulously categorized based on their peptidase family type (MEROPS ID), sequence identity, sequence length, RMSD proximity, and the conformational variations observed in their variable regions (Appendix A). Within the framework of the categorization process, a percent identity matrix was generated via the Clustal Omega server [28], while a matrix of pairwise similarities was constructed using the STAMP program on the SALIGN server [29] (Appendix A). The superposition of these crystal structures distinctly delineated the structurally conserved regions (SCRs) and the variable ones (Figure 1). Our structural analysis revealed a remarkable similarity in structure among penicillin G acylase (PGA) enzymes from *Providencia rettgeri*, *Escherichia coli*, *Kluyvera citrophila*, and *Kluyvera cryocrescens*, suggesting a conserved architectural framework within these species. The penicillin G acylase from these species exhibits lengths ranging from 750 to 818 amino acids. It is important to note that the *Kluyvera cryocrescens* penicillin G acylase reported in the Protein Data Bank (PDB codes: 7REO and 7 REP) has only one chain, and the portion corresponding to the A chain is at the end of the amino acid sequence compared to other enzymes of the same type. In contrast, the PGA enzyme from *Alcaligenes faecalis* displayed distinctive structural features and a length range near 745 amino acids, indicating potential functional divergence compared to other PGA enzymes. Intriguingly, PGA enzymes from various *Bacillus* sp. exhibited a high degree of structural similarity, with lengths ranging from 730 to 719 amino acids. There is a notable correlation between the length of the PGA enzymes and their structural features. Longer PGA enzymes, such as those from *Providencia rettgeri*, *Escherichia coli*, and *Kluyvera citrophila*, exhibited a more extended structural framework, potentially accommodating additional functional domains or regulatory elements. Conversely, shorter PGA enzymes, like those from *Alcaligenes faecalis*, displayed a more compact structure, suggesting a streamlined architecture possibly optimized for specific catalytic activities. This correlation between enzyme length and structural complexity provides further insights into the functional diversity of PGA enzymes across different bacterial species. An interesting observation is the high structural similarity, indicated by low RMSD values, observed in D-succinylase enzymes from *Cupriavidus* sp. P4-10-C (G-), despite having a sequence similarity of only 27% with the other enzymes analyzed (Figure 1). This finding underscores the conservation of key structural elements essential for the enzymatic function of D-succinylase across diverse bacterial species, despite a significant sequence divergence.

Our comparative structural analysis revealed a striking similarity among cephalosporin acylases of *Brevundimonas diminuta*, *Pseudomonas* sp. 130, *Pseudomonas* sp. SY-77-1, and *Pseudomonas* sp. GK16, indicating a conserved structural framework across different bacterial species. However, two enzymes, 4HSR and 4HST, both belonging to the genus Pseudomonas, exhibited notable deviations from this pattern. Unlike other Pseudomonas enzymes with a maximum chain length of 684 amino acids, 4HSR and 4HST possessed longer chains comprising 751 amino acids and remarkable structural differences. The acyl-homoserine lactone acylase enzymes exhibit distinct structural characteristics compared to other N-terminal nucleophile (Ntn) amide-hydrolases S45 family. However, among these enzymes, there are differences in the length of their sequences, ranging from 768 amino acids in tha case of *Acidovorax* sp. MR-S7 to 710 amino acids in *Pseudomonas aeruginosa*, which cause notable structure differences between these two species. An intriguing observation worth noting is that acyl-homoserine lactone acylase enzymes exhibit less than 20% sequence similarity with penicillin G acylase. However, they demonstrate a higher degree of structural resemblance to penicillin G acylase than the structural similarity observed between penicillin G acylase and cephalosporin acylase, despite the latter two sharing slightly over 20% sequence similarity. Despite the low percentages of similarity in their sequences, all the enzymes studied share common structural characteristics and a central structural core in both of their chains (Figure 1B).

#### 2.1.1. Chain A

The first module corresponds to what, in most crystallographic studies reported in the PDB, is termed chain A, and is also referred to as subunit α, which is one of the four structural elements of their respective genes in this protein family (Figure 2). Among the studied sequences, this module was found at the C-terminal end in only four cases, while, in the rest of the proteins, it occupies the first third of the amino acid sequence at the N-terminal end. The initial structurally conserved region (SCR1A) in these proteins commences with a beta hairpin motif. The antiparallel beta-sheets consist of five amino acids each, linked by a turn of four amino acids. This loop features a highly conserved Gly at its N-terminus, preceded by Y, M, W, or F. Subsequent to the hairpin motif, there is a distorted V-shaped alpha-helix (HA) spanning 27 amino acids. The distortion primarily arises from a triplet sequence, RLF, HGD, NAC, NLC, or RLW. HA concludes with a highly conserved Gly, forming a helix–turn–helix motif. The first SCR (SCR1A) culminates in the subsequent helix (HA´), a short helix comprising 1 to 1.5 turns, terminating with another highly conserved Gly at its C-terminus. In most cases, a second helix–turn–helix motif is observed, whereas, in some instances, an additional loop of varying size (ranging from 6 to 16 amino acids) connects the first SCR (SCR1A) to the second SCR (SCR2A). The second SCR (SCR2A) consists solely of a nine-amino acid alpha-helix (HB). SCR2 and SCR3 are linked by a variable region capable of forming an alpha-helix (HB´) of variable size positioned between two loops; the total length of this structurally variable region is 17 amino acids. The third SCR (SCR3A) comprises a long alpha-helix (HC) spanning 20 amino acids, passing beneath HA and forming the protein’s base. The final SCR (SCR4A) is composed of a loop running antiparallel to HC and an α-helix (HD) perpendicular to HC, measuring up to 15 amino acids in length. It is important to note that only the initial six amino acids of HD are considered part of the fourth SCR. The remaining segment within this module cannot be classified as SCR due to its high structural variability. A summary table of the identified SCRs is presented in Appendix A.

#### 2.1.2. Chain B

The second module corresponds to what, in most proteins, is termed as chain B in crystallographic structures (Figure 3). The structure corresponding to chain B initiates with a conserved beta-sheet (B1-1) present in all types of this proteins (SCR1B). The first amino acid in this chain is typically a Ser, reported as being crucial for the protein’s activity [13,30], followed by the sequence NXWXXG. B1-1 consists of approximately seven amino acids and is interrupted by a G, A, or R, followed by a variable region, which presents loops, turns, or even helices depending on each specific protein. This initial SVR spans eight amino acids across all studied structural types. SCR1A continues with an antiparallel beta-sheet (B1-2), mirroring B1-1. Like B1-1, B1-2 also comprises seven amino acids. This region terminates in a highly conserved P in the protein’s sequence (P28). It is worth noting that P28 is part of a consensus sequence of penicillin acylase family proteins in reported bacterial species in NCBI; LLANDHPH. Depending on each protein, a second variable region may extend B1-2 and contains loops or turns spanning eight to nine amino acids. The subsequent SCR (SCR2B) commences with an eight-amino acid beta-sheet (B2-1), followed by a semi-conserved loop of two to three amino acids and continues with an antiparallel beta-sheet (B2-2) also spanning eight amino acids. In B2-1, an HL or LH pattern near its C-terminal end is consistently observed across all sequences studied, both in the three-dimensional structures and sequence alignments, with L always aligning at the same position. In the middle of B2-2, a conserved Gly appears across all sequences. This is followed by a variable region of seven or eight amino acids containing loops or turns that connects to SCR3B. SCR3B consists of a beta hairpin motif comprising a short three-amino acid beta-sheet (B2-3), a three-amino acid coil, and a six-amino acid beta-sheet (B2-4). SCR3B further includes a six-amino acid coil from the active site, and an eight-amino acid beta-sheet (B3-1). In this region, an alanine has been identified as being crucial for the activity of this protein [26]. It is significant that this amino acid is not conserved in the sequences of some studied proteins, possibly varying to V, H, or R. However, threonine is found in all sequences preceding this amino acid. The activity of this threonine has not been reported; yet, its presence across all sequences could be fundamental to the protein’s function.

A short variable region comprising three to six amino acids connects with SCR4B. SCR4B features a brief four-amino acid beta-sheet (B4-1), followed by a turn, and a second four-amino acid beta-sheet (B4-2) interrupted by a coiled distortion involving a single amino acid (usually a long aliphatic residue such as M, L, F, etc.) that connects to the seven-amino acid beta-sheet (B5-1). Despite being classified as an SCR, SCR4B displays low RMSD among the studied structures, suggesting it represents a region of high structural flexibility. Given its proximity to the active site, it may be hypothesized to act as a substrate recognition region. SCR4B and SCR5B are linked by a structurally variable region of six to ten amino acids length. SCR5B forms a hairpin motif composed of the nine-amino acid beta-sheet (B5-2), a turn, and the four-amino acid beta-sheet (B5-3). SCR6B constitutes a single beta-sheet (B3-2). Following this region, there is a lengthy variable region spanning 20 to 21 amino acids. Within this region, the first structurally non-conserved α-helix may be observed. This region holds significance as it contains an amino acid reported to interact with the substrate at the active site [31].

SCR7B comprises an eight-amino acid alpha-helix (H1). H1 connects to SCR8 via a six- or eight-amino acid variable region. SCR8B initiates at its N-terminus with a seven-amino acid beta-sheet (B2-5), followed by a semi-conserved four- or five-amino acid turn, and then the beta-sheet (B2-6). B2-5 exhibits an interesting pattern: a typically hydrophobic amino acid in crystallographic structures (I, F, V, or L, with H present in 4HSR and 4HST) or a conserved Q in PGA bacterial sequences, followed by a highly conserved N. Notably, this N, like the B2-4 threonine, lacks the reported function in the literature. However, its presence across all sequences and interaction through hydrogen bonding with another highly conserved amino acid (B2-4 T) suggests its potential indispensability in the protein’s function. Similar to SCR3, SCR8 extends into a coil and the beta-sheet (B6-1). Subsequently, a region within the solvent-accessible surface area varies considerably in length, ranging from 7 to 14 amino acids. Proteins with increased amino acid counts tend to form alpha-helices in this region. This segment is succeeded by a short four-amino acid beta-sheet (B3-3), denoted as SCR9B. A Gly at the extreme C-terminal of B3-3 is highly conserved, suggesting its indispensability in maintaining the protein structure. Following SCR9B is a structurally variable region of varying length, ranging from 6 to 16 amino acids. In this section, a highly conserved W residue is identified. This segment connects to another short four-amino acid beta-sheet (B6-2), succeeded by a 310 helix and the four-amino acid beta-sheet B2-7. Notably, the 310 helix and beta-sheet B2-7 are linked by a long-sidechain amino acid like M, L, I, or N, bonded to a highly conserved P, initiating B2-7. Termed as SCR10B, this region has a 12-amino acid length; four amino acids from B6-2, three from the 3_10_ helix, the long-sidechain connector amino acid, and four amino acids from the B2-7. SCR3B, SCR8B, and SCR10B serve as the longest conserved regions traversing the entire protein’s globular core structure, establishing its structural framework.

B2-7 is connected to SCR11B by a two- to four-amino acid turn. SCR11B comprises a beta-sheet B (B1-3) and initial segment of a turn in the C-terminal of B1-3. Subsequent to this, there exists a region of variability spanning between 17 to 27 amino acids, succeeded by SCR12B, a nine-amino acid α-helix, denoted as H2. While H2 may span up to 15 amino acids, only 9 amino acids are within an SCR. The constant segment of H2 starts with a highly conserved R, recognized as vital for this protein’s functionality [21]. Within this region and spaced by two amino acids, an E, Q, or R is situated. A variable-length loop (spanning from 8 to 17 amino acids) connects to the subsequent SCR. SCR13B encapsulates eight amino acids from H3, with variable length, potentially extending up to twelve amino acids. From this point onward, a region characterized by high structural variability begins. It is rich in alpha-helices, loops, and few beta-sheets, lacking any clear structural relationship among proteins of this type (Figure 3B). This region may be specific to the requirements of each species and protein. This section encompasses regions that could be regarded as conserved. For instance, there exists a region with a loop–helix–loop–helix–loop motif bearing a highly conserved LXXW sequence that marks the beginning of the first helix. However, the structural alignment (STAMP/fit) does not identify them as conserved regions, as it displays the significant distance among these motifs across different (Ntn) amide-hydrolase S45 types. This region might exhibit high flexibility.

The following SCR (SCR14B) reappears 200 amino acids later. SCR14B consists of the eight-amino acid beta-sheet B1-4. A variable-length region separates B1-4 from B1-5, which forms the penultimate SCR; SCR15B. B1-5 can span up to nine amino acids; however, only four amino acids remain constant in all the studied structures. SCR15B is followed by an variable region formed by a loop of 15 to 20 amino acids to connect to the final SCR. SCR16 represents a distinct region characterized by the presence of the seven-amino acid H4, linked through two amino acids to the four-amino acid B1-6, which is subsequently connected by two amino acids to the six-amino acid H5. A subsequent sequence of 2 to 13 amino acids forms a loop, representing the culmination of these protein structures. At the beginning of this region, there is a conserved DQ duplet, with the presence of D residue observed in all sequences analyzed. A summary table of the identified SCRs is presented in Appendix A.

#### 2.1.3. Active Site

To compare the composition of the elements located within the active site, 10 structures were used as a reference frame (Table 1 and Figure 4). The table represents a comparative analysis of specific amino acid residues across multiple structures (1FXH, 1FM2, 1KEH, 2WYB, 3K3W, 4HSR, 4YF9, 6NVW, 7EA4, and 7REO) of penicillin G acylases (PGA) and related enzymes. The observations focus on the conservation or variation of particular residues within the active site of these enzymes. It highlights conservation patterns, catalytic residues, and key positions concerning structural features and functional aspects. The details include residue positions, their conservation, involvement in catalysis, and locations within structural domains or active sites. Notable residues, such as A-M142, A-R145, A-F146, B-S1, B-P22, B-Q23, and B-R263, are highly conserved among orthologous genes, predominantly near SCR4A and SCR1B; this region is noted for its reported interaction with the substrate at the active site. [32]. These residues play roles in catalysis, ortholog conservation, and structural stability, as reported in various species and crystallographic structures; particularly, A-R145 plays a significant role in binding the β-lactam moiety of penicillin G, extending into the solvent and maintaining a distance of 8 Å from the ligand’s carboxyl group, and A-F146 can potentially be relocated to facilitate the entry of penicillin to the active site [31]. Additionally, residues like B-S1, B-A69, and B-R263 are confirmed catalytic amino acids, involved in critical hydrogen bond interactions and structural co-ordination [30]. Mutations in these amino acids have been reported, where the protein completely lost its intramolecular autoproteolytic activity [33,34,35]. Conversely, residues like B-F24, B-F57, and B-V56 show the variability between species, impacting structural modules and catalytic site features. F24 interacts with residue R145 and with penicillin within the active site. Mutations in these residues may influence the enzyme’s Km. The presence of leucine (L) is more frequent than phenylalanine [21,31]. Further, B-N241 stands out as highly conserved and catalytically essential, notably observed at SCR11B, indicative of its crucial role in enzymatic function across species. The highlighted residues signify diverse functionalities, conservation patterns, and catalytic roles, shedding light on these enzymes’ intricate architecture and functional mechanism [31,36]. A detailed version containing specific observations for each amino acid, as presented in Table 1, is available in the Appendix A. Additionally, a figure illustrating the structural superposition of a segment of the secondary structure within the enzyme cavities can be found in the Appendix A.

Observations and bibliographic revision made on the amino acid residues within the structures of penicillin G acylases (PGAs) and related enzymes confirm significant insights: (1) Catalytic amino acid: Residue Β-S1, positioned at the beginning of SCR1Β, is an amino acid with confirmed catalytic significance. Experimental studies in E. coli affirm its catalytic role, impacting intramolecular autoproteolytic activity. (2) Hydrogen bond network and catalysis: The residues Β-Q23, Β-F57, and Β-R263 are part of the catalytic triad and essential for enzyme function, forming critical hydrogen bonds and co-ordinating the scissile amide bond. Residue Β-Q23 is part of the catalytic triad (Asp-His-Ser) crucial for enzyme function. It plays a significant role in forming hydrogen bonds and co-ordinating the scissile amide bond of PG, essential for the acylenzyme intermediate formation [30]. (3) Structural–functional variability driven by sequence differences: Certain residues, like A-F146, B-Q23, Β-V56, Β-F71, and Β-W154 (in 1FXH), exhibit variability between different species and crystal structures. This observed variance hints at potential divergences in substrate binding specificity or enzymatic activity among various Ntn amide-hydrolases. Β-V56 is a non-conserved residue in the active site; it is found at the beginning of SCR3Β but is not part of it. Similarly, Β-F57 displays hydrophobic variability among I, V, L, M, A, and F residues at the start of SCR3Β. B-W154 is part of a variable region between SCR6Β and SCR7Β and it has been observed to interact with the substrate in binding site (25). Specific residues, such as A-F146 and Β-Q23, play pivotal roles in the functional dynamics of these proteins. A-F146’s potential mobility may facilitate the ingress of PNN (specific environmental factor) [31], while, Β-Q23, a constituent of the catalytic triad [30], exhibits a lower variability, potentially substitutable by histidine; unlike Β-Q23, A-F146 displays a more pronounced variability. The variation in these residues, observed across species and crystal structures, underscores the possible divergence in substrate recognition, specificity in binding, or enzymatic activity across distinct bacterial strains. (4) Conservation among orthologous genes: Residues Β-S1, Β-P22, Β-T68, Β-N241, and Β-R263 show high conservation among orthologous genes. These residues are located at the beginning of SCR1Β, at the end of SCR1Β, near the C-terminus of Β2-4 in SCRΒ3, at SCR11Β, and at the SCR12 near the C-terminus of H2; respectively. (5) Novel findings: Observations like the highly conserved Β-T68 near the C-terminal of Β2-4 in SCR3Β and Β-N178 at the N-terminal of Β2-5 in SCRΒ8, deduced by observation but not previously reported, suggest new structural insights. The residue Β-N178 interacts via hydrogen bonding with Β-T68, implicating a potential functional role, yet previously unreported. There are differences in the active site structure between PDB structures 7REO and 7REP, potentially attributed to mutations at positions G72A and D74S. However, despite these adjustments induced by mutations, no significant effect of the ligand on the overall structure of any enzyme is observed. Moreover, the enhanced structural stability, may play a protective role in the periplasmic environment against structural perturbations. Interestingly, no evidence of induced fit, where the enzyme undergoes conformational changes upon substrate binding, was observed in our analysis. These observations offer a comprehensive understanding of residue conservation, catalytic significance, structural positions, and their relevance in the context of PGA and related enzyme structures, bolstered by references from scientific studies.

##### Variable Regions on the Active Site

There are two variable regions of interest constituting the active site (Figure 3B and Figure 4). In chain A, immediately following SCR4, and in chain B, there is a variable region between SCR6B and SCR7B. Both regions display a combination of alpha-helices and loops. In crystal structure 1FXH, a phenylalanine is found at position 146 in chain A, while a tryptophan is at position 154 in chain B; both are presumed to influence selectivity and activity. Due to its proximity to the active site, this residue has a strong evidence of being involved in the binding of the β-lactam moiety [31]. The discernment of residues linked to these positions would offer advantages in design of enzymes with customized properties. Due to its placement within extensively varied regions, pinpointing this position in enzymes lacking structural data is not achievable solely through sequence alignment. An in-depth examination of these regions within the structures scrutinized in this study unveiled a consistent presence of a specific residue across nearly all compared protein structures. This residue is strategically oriented toward the active site, retaining its stability throughout substrate binding, as evidenced by comparisons among multiple identical protein structures. Through the structural alignment (STAMP/fit) of the protein structures, it was demonstrated that this specific position consistently corresponds to the placement of phenylalanine and tryptophan observed at positions A146 and B154 in the 1FHX structure, respectively (Table 1 and Figure 4). In homology modeling, particular attention must be devoted to these regions to ensure the utmost precision in delineating the active site amino acids of this variable region.

## 3. Discussion

Despite their notable divergence in sequence, these enzymes exhibit a shared structural architecture. The well-defined elements of the secondary structure are consistently observed across all identified crystal structures, encompassing their active site [13,31,37]. A comprehensive structural alignment comprising 51 related structures unveiled conserved structural domains that encapsulate the majority of described secondary structure elements within the framework of penicillin acylases, cephalosporin acylases, and D-succinylase. It was discerned that specific secondary structure elements coalesce into distinct structural units, termed as structurally conserved regions (SCRs) 1–16, underscoring the modular organization inherent in these enzymes. Generating a dependable structure-based profile applied across all instances of penicillin acylases, cephalosporin acylases, and D-succinylase in the Protein Data Bank (PDB) not only facilitated the prediction of conserved regions but also tackled a more formidable challenge: the identification and categorization of variable regions.

Given that residues determining substrate specificity in the N-terminal nucleophile (Ntn) amide-hydrolases S45 family are believed to be situated within the variable regions, and those governing catalytic activity lie within the SCR [13,14], their identification holds significant relevance for biochemical property manipulation. Among the sixteen proposed SCRBs, SCR1B, SCR3B (comprising the motif beta-sheet B2-4-coil-B5-1), and SCR8B, along with the variable regions adjoining SCR6B and SCR7B, and after SCR4A, are situated adjacent to the substrate binding cavity [33,34,38]. Consequently, it is presumed that they potentially interact with the substrate. SCR1B harbors highly conserved residues that were previously acknowledged as essential for activity and are now recognized as pivotal for the three-dimensional conformation of these enzymes [30]. Specifically, the residue at position 22, directly oriented toward the active site, consistently appears as proline across all N-terminal nucleophile (Ntn) amide-hydrolase structures and sequences analyzed. Positioned at residue W154 (in 1FXH) and directly facing the active site, its location within the highly variable loop or helix region poses challenges in its identification within sequences lacking structural data.

The identification of positions corresponding to S1, P22, Q23, and F57 in 1FHX across all structures was accurately anticipated following multiple sequence alignment across the analyzed structures. This prediction was subsequently validated upon examination of more recent structural data. The comprehensive analysis of these positions across all sequences unveiled that residues at position 23 primarily exhibit a polar nature, while those at position 57 are predominantly aliphatic, with occasional occurrences of small polar amino acids and, rarely, charged residues. Given that the residue characteristics at this position significantly impact substrate specificity and regioselectivity, their precise identification aids in the strategic design of enzymes to possess more favorable attributes for biocatalytic applications. The challenge in predicting the structure of these enzymes lies in the extensive variable region between SCR13B and SCR14. This nearly unstructured region demonstrates that areas with the highest structural variability are situated on the protein’s surface. This observation suggests the distinct characteristics of each species, potentially delineating species-specific protein–protein interactions.

The correlation between the sequence identity (Clustal Omega EMBL server [28]) and structure similarity score (SALIGN [29]) (Appendix A) reveals that pairs of crystallographic structures with sequence identity percentages of both 100% and 20% can exhibit similar structural similarity scores. Conversely, pairs of structures with 20% sequence identity may display structural differences ranging from approximately −1.5 to −2.6 on a scale from −3 to 0. These observations underscore the importance of considering factors beyond the sequence similarity percentage when selecting a template for protein modeling. Such considerations should include the presence or absence of ligands, sequence length, and other pertinent factors. Indeed, chain length plays a pivotal role in determining the three-dimensional structure of proteins. The length of the polypeptide chain directly influences the folding patterns and interactions between amino acid residues, ultimately dictating the overall protein conformation. In the context of the N-terminal nucleophile (Ntn) amide-hydrolases S45 family, variations in chain length can lead to structural alterations that may impact their catalytic activity, substrate specificity, and stability. Longer chains may accommodate additional structural elements or functional domains, potentially conferring unique biochemical properties to the enzyme. For example, variability is evident in the lengths of the loops connecting helices within the non-conserved region. Models 1FXH and 7REO exhibit an elongated loop, concomitant with a reduction in the volume of the entrance to the active site, suggesting a potential regulatory role in access. Conversely, models 6NVW and 8BRQ display shorter loop regions between helices, implying a differential influence on active site accessibility, indicative of specific adaptations. However, in the case of the 1GHD model, pertaining to glutaryl-7-aminocephalosporanic, despite featuring a short loop in the aforementioned region, an extended loop is observed between two beta-sheet structures in another region encompassing W440 to P451, occupying the same spatial domain. This may denote a tailored adaptation to restrict access to the active site [39]. Incorporating proline into β-turns is an effective strategy for enhancing protein thermostability by stabilizing loops. Factors such as the preference for Arginine over Lysine, the reduction in thermolabile amino acids, the increase in proline content, and the presence of stable ionic pairs are determining factors in this enhancement. Thermostability, influenced by the three-dimensional structure and amino acid sequence, is affected by elements such as the deamidation of Asparagine and Glutamine at high temperatures [40]. In comparing models from thermophilic organisms (6NVW and 8BRQ), they exhibit a lower abundance of proline, similar to positions in E. coli (1FXH), except in the loop region of SCR8B, which connects two folded beta strands. The observed structural differences between PDB structures 7REO and 7REP underscore the importance of mutations in modulating enzyme conformation. However, the lack of significant alterations in the overall structure suggests robustness and resilience in periplasmic enzymes. Furthermore, the correlation between periplasmic protection and structural stability highlights the adaptation of bacterial enzymes to their microenvironment for optimal functionality and survival. Moreover, the differences in chain length and proline frequency among the N-terminal nucleophile (Ntn) amide-hydrolases S45 family from different bacterial species reflect evolutionary adaptations tailored to their respective environmental niches or biological functions. Understanding the relationship between the chain length and protein structure is, therefore, essential for deciphering the functional diversity and evolutionary history of these enzymes. 

## 4. Materials and Methods

### 4.1. Data

A collection of 83 PDB structures was obtained from the RCSB PDB database [41], as listed in the Appendix A; these represent all crystallographic structures reported up to January 2024. This selection encompasses bacterial structures of penicillin G acylases, cephalosporin acylases, and D-succinylase, all classified under Family S45 in the MEROPS database [5], which includes self-cleaving precursor proteins of N-terminal nucleophile acylases. Notably, structures 1KEC, 3S8R, 4E55, 4E56, and 4E57 exhibit an anomaly with an alanine (A) instead of serine (S) at position 1 of chain B, leading to a lack of autocatalysis and presenting as a single chain. To ensure precise structural alignment, these structures were separated into their two hypothetical chains starting from this Ala1 position. Similarly, structures 5C9I, 7EA4, and 7EBY underwent similar treatment. Additionally, structures 5UBK, 5UBL, 7REP, and 7REO were reversed, beginning with the first amino acids as chain B and ending the last amino acids as chain A. These structures were also processed to present both their hypothetical chain A and chain B for analysis, aligning them with the other structures. These structural adjustments were made for methodological purposes and to ensure quality in both structural and sequence alignments. The inclusion of all crystallographic structures, without distinguishing between wild-type and modified enzymes, enables the investigation of the effects of sequence modifications on protein structure. This approach enhances our understanding of structure–function relationships in these enzymes, facilitates the prediction of more accurate three-dimensional models, and offers a novel method for identifying targets to improve the biocatalytic properties of these enzymes.

### 4.2. Sequences

The examination of N-terminal nucleophile (Ntn) amide-hydrolases S45 family sequences and structures was conducted by comparing them with the putative penicillin acylase [Streptomyces griseus subsp. griseus NBRC 13350] (GenBank: BAG21123.1) [42,43,44]. The top 500 non-redundant matches were identified using the BLAST program (https://blast.ncbi.nlm.nih.gov/Blast.cgi, accessed on 4 January 2024). The sequences were downloaded from the NCBI database. These sequences were annotated based on automatically extracted GenBank annotations [45]. Additionally, secondary structure information was obtained from DSSP annotations within the multisequence alignments for homologous families that included members with available PDB structures. A multiple sequence alignment of all non-redundant and PDB structure sequences was generated using the sequence analysis tools provided by EMBL-EBI, specifically Clustal Omega [28], and manually refined using BioEdit as necessary.

### 4.3. Structural Analysis

SCRs were identified through the creation of a structure-based multisequence alignment using the MultiSeq tool within the VMD software 1.9.3 [46] program for aligning protein molecules, which operates based on a three-dimensional structure. STAMP employs a method that minimizes the Ca distance between aligned residues of each molecule by applying globally optimal rigid-body rotations and translations. It is important to note that alignments can only be performed on structurally similar molecules; attempting to align proteins lacking common structures will render STAMP unable to align them. STAMP assesses the probability of structural equivalence of residues [47] and employs the Smith–Waterman algorithm [48] to determine the best path through a matrix of numerical pairwise similarity values of corresponding sequence positions. This allows STAMP to calculate two alignment confidence measures: P’ij, a measure for residue equivalence, and Sc, the STAMP score, which reflects overall alignment quality. An Sc > 5.5 indicates a high degree of similarity between the considered structures. Regions with P’ij > 6.0 denote conserved secondary structures. To incorporate secondary structure information, STAMP utilizes DSSP outputs [49]. These data can be exported to a FASTA file using the MultiSeq tool. Regions with high P’ij values, indicative of conserved secondary structures, were designated as SCRs, extracted from the alignment, and visualized using VMD [46]. MultiSeq utilizes the STAMP [50], Discovery Studio, and Chimera [51], with the structure from PGA PDB code 1FXH [31] as a reference. Additionally, to assess structural similarity, the SALIGN server was employed [29]. This web server automatically selects the best alignment procedure based on the provided inputs. Multiple alignments are guided by a dendrogram computed from a matrix of all pairwise alignment scores. When aligning sequences to structures, SALIGN utilizes structural environment information to optimize gap placement. If two multiple sequence alignments of related proteins are provided, the server conducts a profile–profile alignment.

## 5. Conclusions

In this study, we employed a methodology to analyze bacterial N-terminal nucleophile (Ntn) amide-hydrolase sequences, enabling the identification of functionally significant residues, structurally conserved regions, and functionally relevant sites within variable regions. Through this approach, we uncovered sixteen structurally conserved regions (SCRs) and key substrate binding sites, which form the structural core of these family enzymes, marking the structural core of these proteins. The identification of these regions, crucial for substrate interaction, was achieved solely through sequence analysis, bypassing the need for structural data. The recognition and interaction regions crucial for substrate binding were delineated without a reliance on structural data, solely leveraging sequence information. Structural predictions were generated for sequences akin to these enzymes or suspected to share similar folding patterns. This navigation approach within bacterial N-terminal nucleophile (Ntn) amide-hydrolases S45 family sequences, pinpointing functionally pertinent sites, offers a distinct advantage in identifying promising targets for the enhancement of these enzymes’ biocatalytic properties. Furthermore, the results enable the creation of more precise three-dimensional models for various penicillin acylases, cephalosporin acylases, and D-succinylase. Importantly, the approach utilized in this case is generic and can be applied to other enzyme families, facilitating the discovery of conserved functional regions and enhancing our understanding of enzyme mechanisms across various biological systems. 

## Figures and Tables

**Figure 1 ijms-25-06850-f001:**
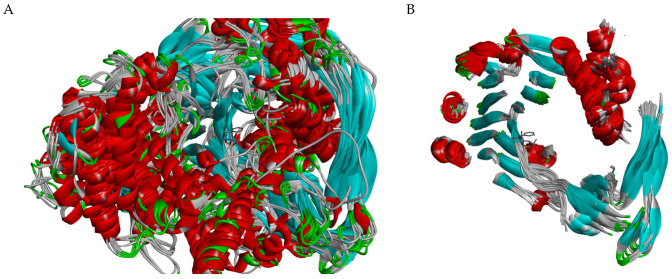
Structural alignment of N-terminal nucleophile (Ntn) amide-hydrolases S45 family. (**A**) The structural alignment of two proteins of each group, specifically referencing entries 1AI4, 1CP9, 7REO, 7REP, 3K3W, 3ML0, 6NVW, 8BRR, 7EA4, 7EBY, 4HSR, 4HST, 4YF9, 5C9I, 3SRA, 5UBK, 1GK1, and 1OR0. (**B**) The structural alignment of SCRs of two proteins of each group, specifically referencing entries 1AI4, 1CP9, 7REO, 7REP, 3K3W, 3ML0, 6NVW, 8BRR, 7EA4, 7EBY, 4HSR, 4HST, 4YF9, 5C9I, 3SRA, 5UBK, 1GK1, and 1OR0. α-helices are marked in red; β-sheets are marked in cyan; loops are marked in white; turns are marked in green; and penicillin is depicted in stick representation for reference [PDB code: 1FXV].

**Figure 2 ijms-25-06850-f002:**
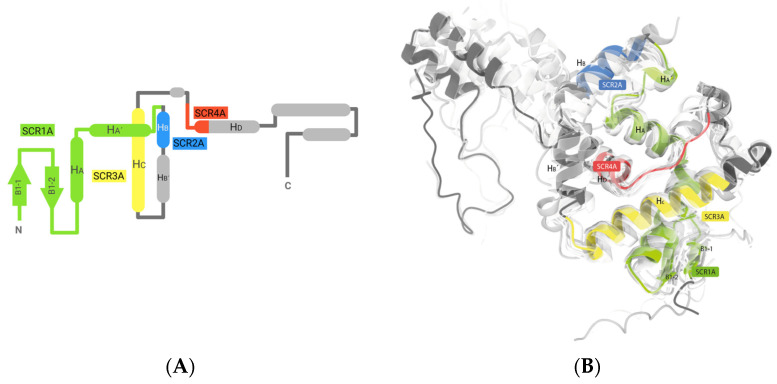
SCRs of the chain A of N-terminal nucleophile (Ntn) amide-hydrolases S45 family are depicted both in a structural overview and in a topological illustration. (**A**) Structurally conserved regions (SCRs) of chain A in penicillin acylases, cephalosporin acylases, and D-succinylase using a topological representation. These conserved regions were identified through STAMP alignment, with their positions highlighted in distinct colors. α-helices are depicted as bold lines, while β-sheets are represented by arrows. (**B**) This provides a structural overview of the structurally conserved regions (SCRs) of chain A in penicillin acylases, cephalosporin acylases, and D-succinylase. These conserved regions, identified through STAMP alignment, are superimposed on the reference structure of penicillin acylase from *Escherichia coli* [PDB code: 1FXV]. The SCRs are depicted in various colors, while the variable regions are represented in grey.

**Figure 3 ijms-25-06850-f003:**
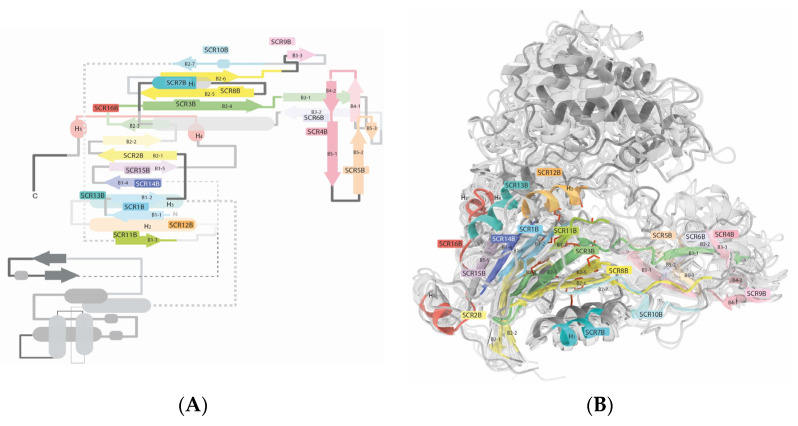
SCRs of the chain B of N-terminal nucleophile (Ntn) amide-hydrolases S45 family are depicted both in a structural overview and in a topological illustration. (**A**) Structurally conserved regions (SCRs) of chain B in penicillin acylases, cephalosporin acylases, and D-succinylase using a topological representation. These conserved regions were identified through STAMP alignment, with their positions highlighted in distinct colors. α-helices are depicted as bold lines, while arrows represent β-sheets. (**B**) This provides a structural overview and structural alignment of penicillin acylases, cephalosporin acylases, and D-succinylase. Structurally conserved regions are depicted in colors, while the variable regions are represented in grey. The most extensive variable region is depicted at the top of the image.

**Figure 4 ijms-25-06850-f004:**
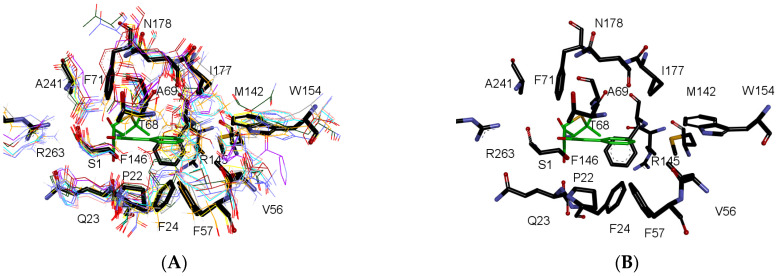
Active site of N-terminal nucleophile (Ntn) amide-hydrolases S45 family. (**A**) The active site overlay of PDB structures of N-terminal nucleophile (Ntn) amide-hydrolases S45 family from various sources is depicted as follows: *E. coli* penicillin G acylase [PDB code: 1FXH] is represented in black sticks for reference, *Pseudomonas aeruginosa* acyl-homoserine lactone acylase [PDB code: 2WYB] in blue lines, *Bacillus megaterium* penicillin G acylase [PDB code: 6NVW] in orange lines, *Kluyvera cryocrescens* penicillin G acylase [PDB code: 7REO] in yellow lines, *Brevundimonas diminuta* cephalosporin C acylase [PDB code: 1KEH] in pink lines, *Acidovorax* sp. MR-S7 acyl-homoserine lactone acylase [PDB code: 4YF9] in green lines, *Cupriavidus* sp. P4-10-C D-succinylase [PDB code: 7EA4] in red lines, *Brevundimonas diminuta* cephalosporin C acylase [PDB code: 1FM2] in cyan lines, and *Pseudomonas* cephalosporin acylase [PDB code: 4HSR] in violet lines. Penicillin is represented in green stick configuration as a point of reference [PDB code: 1FXV]. The annotated amino acids correspond to those in *E. coli* penicillin G acylase [PDB code: 1FXH]. Amino acids corresponding to spatial positions in other crystallographic structures are detailed in Table 1. (**B**) The amino acids identified as constituents of the active site in the N-terminal nucleophile (Ntn) amide-hydrolases S45 family in this study are depicted in black sticks. The annotated amino acids correspond to those present in *E. coli* penicillin G acylase [PDB code: 1FXH]. Panels A and B are shown from the same perspective. (**C**) The first line corresponds to the consensus numbering of 83 crystal structure sequences. Residue and chain IDs of the crystallographic structures are indicated on the sides of the alignment. Active site residues are denoted by an asterisk (*). Structurally conserved regions (SCRs) are marked by a hyphen (-) and a vertical bar (|) at the start and at the end. The "#" in "SCR#A" and "SCR#B" is used as a placeholder to indicate that any number can be applied, generalizing the reference to structurally conserved regions in chain A and chain B, respectively.

**Table 1 ijms-25-06850-t001:** The positions corresponding to amino acids within the active sites in structures 1FXH, 1FM2, 1KEH, 2WYB, 3K3W, 4HSR, 4YF9, 6NVW, 7EA4, and 7REO. Amino acids in the same row correspond to those that occupy the same space in the structural alignment and not to those that are aligned in the multiple sequence alignment. In bold letters are shown the residues in direct interaction with penicillin in 1FXV.

1FXH	1FM2	1KEH	2WYB	3K3W	4HSR	4YF9	6NVW	7EA4	7REO
A-M142	A-M145	A-M145	A-L146	A-M143	A-G162	A-N160	A-M145	A-G181	A-M697
A-R145	A-L148	A-L148	A-E149	A-R146	A-L164	A-A162	A-Y147	No fit	A-R700
A-F146	A-Y149	A-Y149	A-G150	A-F147	A-M165	A-G163	A-F148	A-E182	A-F701
**Β-S1**	**Β-S170**	**Β-A170**	**Β-S1**	**Β-S1**	**Β-S1**	**Β-S1**	**Β-S1**	**Β-S282**	**Β-S1**
Β-P22	Β-P191	Β-P191	Β-P22	Β-P22	Β-P22	Β-P22	Β-P22	Β-P303	Β-P22
**Β-Q23**	**Β-H192**	**Β-H192**	**Β-H23**	**Β-Q23**	**Β-H23**	**Β-H23**	**Β-Q23**	**Β-H304**	**Β-Q23**
Β-F24	Β-L193	Β-L193	Β-F24	Β-F24	Β-R24	Β-W24	Β-V24	Β-R305	Β-F24
Β-V56	Β-R226	Β-R226	Β-N57	Β-L56	Β-P56	Β-Q57	Β-M56	Β-S337	Β-L56
Β-F57	Β-F227	Β-F227	Β-I58	Β-F57	Β-F57	Β-I58	Β-F57	Β-I338	Β-F57
Β-T68	Β-T238	Β-T238	Β-T69	Β-T69	Β-T69	Β-T69	Β-T68	Β-T349	Β-T68
**Β-A69**	**Β-V239**	**Β-V239**	**Β-V70**	**Β-A70**	**Β-H70**	**Β-V70**	**Β-A69**	**Β-R350**	**Β-A69**
Β-F71	Β-G241	B-G241	B-T72	B-P72	B-F72	B-T72	B-Y71	B-Y352	B-A71
Β-W154	Β-Y322	Β-Y322	Β-W162	Β-W154	Β-L154	Β-W165	Β-Y158	Β-S436	Β-W154
Β-I177	Β-F346	Β-F346	Β-V187	Β-I177	Β-H178	Β-V190	Β-L181	Β-E460	Β-I177
Β-N178	Β-N347	Β-N347	Β-N189	Β-N178	Β-N179	Β-N191	Β-N182	Β-N461	Β-N178
**Β-A241**	**Β-N413**	**Β-N413**	**Β-N269**	**Β-N241**	**Β-N242**	**Β-N278**	**Β-N245**	**Β-N523**	**Β-N241**
**Β-R263**	**Β-R443**	**Β-R443**	**Β-R297**	**Β-R261**	**Β-R263**	**Β-R308**	**Β-R266**	**Β-R547**	**Β-R263**

## Data Availability

https://figshare.com/articles/dataset/PGA_data/25996771 (accessed on 6 June 2024).

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
