# Peer review of "Exploring the Structurally Conserved Regions and Functional Significance in Bacterial N-Terminal Nucleophile (Ntn) Amide-Hydrolases"

_ijms, 2024, doi:10.3390/ijms25136850_

Round 1
Reviewer 1 Report
Comments and Suggestions for Authors
The manuscript was overall well-written, a further language polishing will increase the readability.
Author Response
Dear reviewer
In the attached files, please find a revised version of our manuscript “Exploring the Structurally Conserved Regions and Functional Significance in Bacterial N-terminal Nucleophile (Ntn) Amide-hydrolases” (ijms-3031591). In this letter we are answering the comments of the reviewer.
We thank you and the reviewer for your careful evaluation of our manuscript, and trust the corrections are satisfactory. The suggested revisions have been implemented as follows:
Comment:
- Please read through the whole manuscript to polish the language. There are some minor language issues: such as,
1) In the Abstract, Page 1, Line 17, ‘ … and D-Succinylase’, D-Succinylase is not an antibiotic chemical. Therefore, it should be deleted.
2) In Line 134, …(fig 1a)… should be ..,(Figure 1A)… to be consistent with the figure legend.
3) In line 312 … Escherichia coli… should be italicized.
Answer:
Minor language issues were corrected in lines 17 and 312, and all nomenclature-related issues in the figures were addressed. The species names has been italicized.
Comment
The format for Figure 2 and Figure 3 should be consistent. In Figure 3, Panel B and Panel C are repetitive, panel B can be removed. In Figure 2, the authors show two colored structures, to be consistent with Panel C in Figure 3, please show one structure with color. Please also remove the electron density background in the figure.
Answer:
The format of Figures 2 and 3 has been improved for consistency. Panel B was removed, and the electron density background was eliminated from the figures.
Comments:
The current organization of Table 1 is not easy to follow. The description of observations should not be moved to a proper place in the context. Table 1 might be replaced with a sequence alignment figure with highlighted active site residues.
Figure 4 shows well-colored PDB codes, but the amino acid residues were not colored appropriately. The amino residues might be colored accordingly, which would be much better.
Answer:
Table 1 was modified for clarity. The "Observation" column was removed; however, the table was retained as it is fundamental for understanding Figure 4 and demonstrating the structural alignment of the crystallographic structures. Specifically, it highlights that while certain amino acids do not align in a multiple sequence alignment, they occupy the same spatial position when performing the structural alignment. To complement this information, an image of the multiple sequence alignment was added as suggested. In Figure 4, along with the depiction of the Multiple Sequence Alignment (MSA), we include two additional images showcasing the active site amino acids. One image displays the superposition of active site amino acids from representative crystallographic structures of these enzymes, while the other highlights the active site amino acids from a single structure. We are receptive to retaining both images or selecting only one of the two options, for your deliberation.
Comment:
To increase the significance of this work, include a sentence in the conclusion highlighting how the approach utilized for the particular case analysis is generic.
Answer:
To increase the significance of this work, we have added the following sentence to the conclusion: “Importantly, the approach utilized in this case is generic and can be applied to other enzyme families, facilitating the discovery of conserved functional regions and enhancing our understanding of enzyme mechanisms across various biological systems.”

Reviewer 2 Report
Comments and Suggestions for Authors
comments are in the attached pdf file

Author Response
Dear reviewer
In the attached files, please find a revised version of our manuscript “Exploring the Structurally Conserved Regions and Functional Significance in Bacterial N-terminal Nucleophile (Ntn) Amide-hydrolases” (ijms-3031591). In this letter we are answering the comments of the reviewer.
We thank you and the reviewer for your careful evaluation of our manuscript, and trust the corrections are satisfactory. The suggested revisions have been implemented as follows:
Comment:
“The first module corresponds to what, in most proteins, is termed as chain A (Fig2)”
The name of the chain is automatically associated to the protein structure during crystallographic refinement – there is no rule for it – some crystallographic assemblies can be described by chain A, B, C and D and some by K, L, M, N, etc. While describing a quaternary protein structure you should rather use universal names. Isn’t a penicillin acylase a heterodimer built by two subunits α and β (as other Ntn-hydrolases)? So the fold should be described as αββα heterodimer or (αβ)2 heterodimer, not as chain A and B (it should be changed). You can look e.g. here:
https://www.tandfonline.com/doi/full/10.3109/07388551.2014.960359
https://www.sciencedirect.com/science/article/pii/S0022283601950430
https://www.ncbi.nlm.nih.gov/pmc/articles/PMC2144523/
https://www.ncbi.nlm.nih.gov/pmc/articles/PMC9140057/
Answer:
We have clarified the distinction between crystallographic studies and the nomenclature of the gene. The previous text suggested that the nomenclature of chain A was related to the protein itself rather than the crystallographic study reported. We have corrected this and added several lines in the introduction concerning the synthesis and maturation process of this protein family. It should be noted that the authors were already familiar with the nomenclature and structural characteristics of these proteins. Three out of the four bibliographical references were already cited, and we have now included the recommended reference, among others, to clarify these distinctions and provide a more detailed explanation of the structural nature of these proteins.
Inserted lines in the introduction: “The genes in this protein family encode a precursor polypeptide with four structural elements: a signal peptide, α- and β-subunits, and an inter-subunit spacer. The maturation of the zymogen involves the autocatalytic cleavage of the inter-subunit spacer. The mature protein forms a heterodimer consisting of α and β subunits. The α-subunit folds first and serves as a template for the proper folding of the β-subunit. The first module corresponds to what, in most crystallographic studies reported in the PDB, is termed chain A, and is also referred to as subunit α, which is one of the four structural elements of their respective genes in this protein family.”
Comment:
“It is important to note Kluyvera cryocrescens penicillin G acylase reported in the Protein Data Bank (PDB: 7REO and 7 REP) has only one chain and, the portion corresponding to the A chain is at the end of the amino acid sequence compared to other enzymes of the same type.”
What does it mean that structure 7reo has only one chain? Is it uncleaved precursor of Ntn-hydrolase or is it an engineered protein? The paper (below) said that it was rather genetic manipulation called “circular permutation” and I’m not sure that there structure should be analyzed here as you are analyzing sequences of natural proteins
https://www.science.org/doi/10.1126/science.abn2009?url_ver=Z39.88-2003&rfr_id=ori:rid:crossref.org&rfr_dat=cr_pub%20%200pubmed
https://pubs.acs.org/doi/10.1021/cb300232n
Answer:
We specified that the proposed study does not distinguish between natural and engineered proteins. It focuses on structural insights for engineering these enzymes and enhancing their biocatalytic properties. Crystallographic studies on genetically manipulated proteins support this objective. Despite these modifications, the proteins retained structurally conserved regions, allowing us to understand the impact of sequence manipulation on protein structure. Genetic modifications can help identify crucial functional regions, as sequence alterations highlight important areas for enzyme function. This approach provides a detailed view of the relationship between sequence, structure, and function. Our analysis showed that the modifications did not introduce artifacts in the structurally conserved regions (SCRs) of the proteins but helped verify the central core common to all proteins in this family, both natural and genetically modified.
The following lines were added to the methodology: “The inclusion of all crystallographic structures, without distinguishing between wild-type and modified enzymes, enables the investigation of the effects of sequence modifications on protein structure. This approach enhances our understanding of structure-function relationships in these enzymes, facilitates the prediction of more accurate three-dimensional models, and offers a novel method for identifying targets to improve the biocatalytic properties of these enzymes.”
Comment:
Figure 1 is very difficult to understand – there is too much information and too many panels – figure should be changed. Enzymes form different organisms should have different colors, important residues should be numbered; what is green, red and cyan color (it should be explained)? What residues are presented as a sticks? Some less important panels should be moved to supplementary data.
Answer:
Less significant panels have been relocated to supplementary data. Furthermore, we have delineated the colored structures and included the following sentence in the figure caption: “α-helix are marked in red; β-sheet are marked in cyan; loops are marked in white; turns are marked in green; Penicillin is depicted in stick representation for reference [PDB code: 1FXV]”
Comments:
Figure 2 is OK but too big to and should be 2-times smaller, the same for Figure 3 – as they are now, it is difficult to compare topology diagram with the crystal structure presented in panel A;
Figure 4 is too big and has poor quality and shows almost the same information as Table 1. So I recommend to move Table 1 to supplementary data (it is too big for main text) and correct dimensions and quality of Figure 4. In Figure 4, along with the depiction of the Multiple Sequence Alignment (MSA), we include two additional images showcasing the active site amino acids. One image displays the superposition of active site amino acids from representative crystallographic structures of these enzymes, while the other highlights the active site amino acids from a single structure. We are receptive to retaining both images or selecting only one of the two options, for your deliberation.
Answer:
Figures 2, 3, and 4 were resized and optimized, and the table was revised in accordance with the reviewers' comments.
Comment:
Line 428: In crystal STRUCTURE 1FXH …
Answer:
We added the word “structure” in line 428
Comment:
Line 526: “was obtained from the RCSB PDB database” - PBD is a DATABANK not “database” !!!
Answer: The term "Protein Data Bank" originated during the 1970s with the establishment of PDB as a repository for protein structures. "Database" specifically denotes an organized collection of data that can be accessed, managed, and updated efficiently. RCSB PDB provides access to a diverse array of data concerning protein structures, encompassing information about the structures themselves, sequences, annotations, and more. While "databank" could be an appropriate term, "database" conveys a broader scope, encompassing various data and services, such as search tools, visualization, analysis, and additional functionalities beyond the mere storage of protein structures. The official RCSB PDB server refers to itself as follow: “The RCSB PDB database can now be browsed using this system. Select the ATC tab from the Browse Database interface to navigate through the drug classification hierarchy, view the number of associated PDB structures, and access the related entries.”
https://www.rcsb.org/news/5764422199cccf72e74ca328

Round 2
Reviewer 2 Report
Comments and Suggestions for Authors
Manuscript can be accepted in present form.
Author Response
Thank you very much for the previous comments and for the current "Comments and Suggestions for Authors: Manuscript can be accepted in present form."
We appreciate your thorough review and the valuable feedback provided, which have greatly contributed to the improvement of our manuscript. We are pleased to hear that the manuscript is now considered acceptable in its present form